# Sleep Quality and Perceived Stress among Health Science Students during Online Education—A Single Institution Study

**DOI:** 10.3390/healthcare12010075

**Published:** 2023-12-29

**Authors:** Flóra Busa, Melinda Petőné Csima, Johanna Andrea Márton, Nóra Rozmann, Attila András Pandur, Luca Anna Ferkai, Krisztina Deutsch, Árpád Kovács, Dávid Sipos

**Affiliations:** 1Department of Medical Imaging, Faculty of Health Sciences, University of Pécs, Vörösmarty Street 4, 7621 Pẻcs, Hungary; bflora0711@gmail.com (F.B.); petone.csima.melinda@uni-mate.hu (M.P.C.); kovacsarpad@med.unideb.hu (Á.K.); 2Institute of Education, Hungarian University of Agriculture and Life Sciences, Guba Sándor Street 40, 7400 Kaposvár, Hungary; 3Faculty of Health Science, Doctoral School of Health Sciences, University of Pẻcs, Vörösmarty Street 4, 7621 Pẻcs, Hungary; mjohanna93@gmail.com (J.A.M.); nora.rozmann@etk.pte.hu (N.R.); 4Faculty of Health Sciences, Institute of Emergency Care, Pedagogy of Health and Nursing Sciences, University of Pécs, Vörösmarty Street 4, 7621 Pécs, Hungary; luca.ferkai@etk.pte.hu (L.A.F.); krisztina.deutsch@etk.pte.hu (K.D.); 5Department of Oxyology, Faculty of Health Sciences, Emergency Care, University of Pécs, 7621 Pécs, Hungary; pandur.attila@pte.hu; 6Department of Oncoradiology, Faculty of Medicine, University of Debrecen, Nagyerdei 98, 4032 Debrecen, Hungary

**Keywords:** sleep quality, perceived stress, Athens Insomnia Scale, Perceived Stress Scale, student, smart device, online education

## Abstract

Recently, online education has been gaining prominence in university life. Our survey aimed to examine sleep quality and perceived stress levels among students at the University of Pécs Faculty of Health Sciences. A cross-sectional, quantitative, descriptive survey was conducted between February and March 2023. The online survey included the Hungarian versions of the internationally validated Athens Insomnia Scale (AIS) and Perceived Stress Scale (PSS). Statistical analysis involved descriptive statistics, independent t-tests, analysis of variance (ANOVA), and Mann–Whitney and Kruskal–Wallis tests (*p* < 0.05). We analyzed 304 responses, and females dominated (n = 270; 88.8%). Students in a relationship had significantly higher AIS scores (t = −2.470; *p* = 0.014). Medium average (2.50–3.49) students and those who rarely/never exercise showed significantly higher AIS and PSS (*p* ≤ 0.05). Students on the phone/watching a series during online education, daily laptop/TV use for more than 2 h, and pre-sleep use of smart devices for more than 60 min also negatively affected AIS and PSS scores (*p* ≤ 0.05). Nursing, physiotherapy, and radiography students were the most affected regarding insomnia and perceived stress (*p* ≤ 0.05). Our survey shows that excessive smart device use and lack of exercise are associated with higher stress levels and poorer sleep quality.

## 1. Introduction

Sleep hygiene is the combination of behavioral and environmental factors to improve sleep quality. Most recommendations focus on factors such as caffeine consumption, smoking, duration of physical activity, stress, and nighttime light and noise conditions [1]. It is important to emphasize that these factors, which negatively impact sleep quality, should be avoided by the general population, not just those with insomnia. Sleep problems contribute to the development of cardiovascular diseases, mental health issues, and the formation of cancerous conditions [1,2].

University students often experience suboptimal sleep quality, characterized by irregular sleep patterns and insufficient durations. The demands of academic life, including exams and assignments, contribute to elevated stress levels, potentially disrupting the natural sleep–wake cycle [1,2]. The pervasive use of electronic devices, particularly before bedtime, exposes students to disruptive blue light, further compromising their sleep. Social and extracurricular activities, coupled with the freedom to set individual schedules, often result in inconsistent bedtime routines. Excessive caffeine and stimulant consumption, common among students, can lead to difficulty falling asleep and disrupted sleep architecture. Additionally, the prevalence of mental health challenges, such as anxiety and depression, further exacerbates sleep disturbances in this demographic [3,4].

The phases of sleep can be divided into two major groups (the non-rapid eye movement (NREM) phase and the rapid eye movement (REM) phase). Pre-sleep alcohol consumption prolongs the NREM phase and shortens the REM stage, especially in the second half of the night. An increased likelihood of snoring and obstructive sleep apnea reduce blood saturation levels and, along with polyuria, shorten sleep continuity. Alcohol reduces melatonin levels and dampens the normal circadian fluctuations in core body temperature, affecting the circadian rhythm [5,6].

Internet usage has increased globally in the 21st century, providing better opportunities for education, administration, information retrieval, and communication. During the COVID-19 pandemic, the forced increase in daily internet usage duration heightened the risk of developing dependency among university students. Pre-sleep smartphone usage leads to a deterioration in sleep quality and atrophy of the gray matter in our brains, resulting in weaker learning abilities and concentration and poorer memory [7,8].

Physical inactivity resulting from internet addiction has negative implications for both physical and mental health, influencing our daily activities, such as increasing neglect of household chores [7]. The risk of developing obesity significantly increases due to the omission of regular physical activity and a sedentary lifestyle and is becoming a public health concern. Additionally, the idealized lean, thin image of women and the muscular image of men idealized by Western societies lead to self-esteem issues among adolescents [9,10].

In addition to external factors, mental health also affects sleep quality, and bidirectional relationship is established. Among university students, one of the primary sources of stressors is the university itself. The disruption of students’ psychological well-being (anxiety, depression) is often caused by high expectations from teachers and parents, competition with peers, the frequency of exams, and the large amount of study material [11,12].

Perceived stress among university students is a widely researched phenomenon, often assessed through validated psychological instruments. High academic demands, including exams and assignments, contribute significantly to elevated perceived stress levels [12]. Factors such as uncertainty about the future, financial pressures, and social challenges also play roles in shaping students’ subjective experiences of stress. The impact of perceived stress on mental health and academic performance underscores the importance of targeted interventions and support services for this population [11,12,13].

In recent times, online education has been gaining prominence in university life, extending the time students spend using smart devices. The impact of online education on sleep quality and perceived stress among university students can vary depending on individual experiences, circumstances, and the specific nature of the online learning environment. Several factors can influence how online education affects sleep and stress levels such as isolation and lack of social interactions, increased screen time, technological challenges, and the home environment [7,10].

In our country, citing the economic situation caused by inflation and the war crisis, in February and March 2023, online education took place for a period of 8 weeks. For students, the concept of online teaching became familiar during the waves of the coronavirus, and experiences varied, with some having a positive while others having a negative impact. Therefore, our survey aimed to examine the sleep quality and perceived stress levels during online education among students at the University of Pécs Faculty of Health Sciences.

## 2. Materials and Methods

Cross-sectional, quantitative, descriptive data collection was carried out. We included in our survey students with active student status who were studying in the following programs at our university: radiography, laboratory science, health visitor, physiotherapy, nursing, and emergency medical services. The data collection did not involve questions to identify respondents, ensuring their anonymity. Each respondent was initially informed about the survey process, and a consent statement was also filled out. Completing the questionnaire was voluntary, and respondents could discontinue at any point during the survey.

The survey was conducted between February and March 2023 among students of the Faculty of Health Sciences at the University of Pécs. Our questionnaire was distributed to each student via email in the form of a newsletter sent by the University of Pécs Faculty of Health Sciences. Also, the questionnaire was shared in relevant social media groups of the mentioned specialties.

### 2.1. Measuring Tools

The first section of the online questionnaire contained sociodemographic information such as age, gender, field of study, and the previous semester’s grade point average. We also conducted questions regarding physical activities. We also assessed students’ mobile phone and internet usage habits, asking about daily internet use, time spent watching movies, computer use, and phone use during lectures. Additionally, we inquired about sleep hygiene habits, including activities before bedtime.

We used the Hungarian-validated version of the Athens Insomnia Scale (AIS), which is freely available and is a self-assessment questionnaire to assess sleep disorders. The scale measures the level and severity of sleep disorders, like insomnia. The AIS aims to evaluate individuals’ overall sleep quality and sleep difficulties. The scoring system allows for comparing results and determining the severity of sleep disorders. The questionnaire consists of 8 questions, each with 4 answer options. Responses are scored from 0 to 3, resulting in a total possible score of 24. Four major groups can be defined based on the severity of sleep disorders. Individuals with a total score between 0 and 5 are considered to have minimal or no sleep disorders. Those experiencing mild sleep disorders (6–10) typically face periodic issues. Moderate insomnia (11–15) already affects daily life. A total score between 16 and 24 indicates severe sleep problems impacting physical and mental health [14].

Sauder et al. [15] validated the Hungarian version of the Perceived Stress Scale (PSS). With this freely available scale, individuals can evaluate how stressed they feel daily. The PSS is a self-assessment questionnaire consisting of 10 questions designed to measure the level of emotional stress based on situations and reactions experienced in the past month. By aggregating the results, the PSS is suitable for estimating an individual’s perceived stress level. The questions are scored on a scale ranging from 0 to 4. Three groups are distinguished based on the total score: below 19 indicates a low stress level, between 19 and 32 suggests a moderate level, and above 32 indicates a high stress level [15].

### 2.2. Statistical Analysis

We utilized Statistical Package for the Social Sciences (SPSS) version 23.0 during the statistical analyses. Descriptive statistics, independent sample t-tests, analysis of variance, correlation, and linear regression were employed for data analysis. In the case of non-parametric distribution, the Kruskal–Wallis test was used. The results of the statistical tests were considered significant at a 95% confidence interval (*p* ≤ 0.05).

### 2.3. Ethical Consideration

Our survey received approval from the Regional Research Ethics Committee (9633—PTE 2023).

## 3. Results

### 3.1. Sociodemographic Results

There were 2012 active students in the Faculty of Health Sciences at the University of Pécs during the survey. After removing irrelevant data, only 304 student responses were used for the analysis (response rate 15.10%). Each response is a student enrolled in the undergraduate programs in the Faculty of Health Sciences at the University of Pécs, specializing in radiography (16.8%), laboratory science (8.2%), nursing (17.4%), physiotherapy (39.8%), health visitor (7.9%), or paramedics (9.9%). The vast majority of respondents (88.8%; n = 270) were female. The proportion of those in a relationship (55.3%) and those who are single (44.7%) was nearly equal (Table 1).

First-year students were most willing to complete the questionnaire (37.2%). We defined two major categories based on the previous semester’s average. Students in the larger group (n = 269; 88.5%) had a semester average above 3.5. Regarding the frequency of physical activity, 54.3% of respondents engage in regular sports, while the proportion of those who do not devote time to physical activity in their leisure time is low (5.3%) (Table 1).

A total of 52% (n = 158) of respondents believed that online education is suitable for theoretical learning, but at the same time, 74.3% (n = 226) indicated that their attention was often diverted (Table 1).

The majority (n = 113; 37.2%) spend 1–2 h watching TV daily. Before going to sleep, 39.8% of the respondents (n = 121) spend more than 30 min browsing the internet. A total of 84.5% of the sample (n = 257) spend more than 2 h daily using smart devices. A total of 46.4% (n = 141) rarely read before bedtime, 41.1% (n = 125) of the respondents consume coffee regularly, and 85.5% of the sample (n = 260) do not smoke (Table 2).

### 3.2. Statistical Connections between Athens Insomnia Scale and Perceived Stress Scale Results

Students achieved an average score of 5.91 (SD = 3.79) on the Athens Insomnia Scale, with a reliability of 0.874.

Most students had no sleep disorder (n = 147 (48.4%); AIS 2.80 (SD = 1.63)). A total of 118 (38.8%) students had a mild sleep disorder (AIS 7.50 (SD = 1.40)). Moderate insomnia was represented among thirty-five (11.5%) students (AIS 12.68 (SD = 0.96)), and only three (1.0%) students reported severe sleep problems (AIS 17.11 (SD = 1.01)) (Table 2).

The average score on the Perceived Stress Scale was 28.31 (SD = 10.35), with a reliability of 0.910.

Most students experienced moderate stress levels (n = 126; 41.45%; PSS 25.96 (SD = 3.25)). High stress levels closely follow, constituting 36.84% of the sample (n = 112; PSS 39.29 (SD = 4.93)), while low stress levels account for 21.71% (n = 66; PSS 14.5 (SD = 3.94)) (Table 3).

**Table 2 healthcare-12-00075-t002:** The relationship between students’ smart device use, sleep habits, coffee and smoking habits, and the quality of sleep and perceived stress (** analysis of variance; *** Kruskal–Wallis).

	n	%	AIS	PSS
Spending time daily watching TV or watching movies				
max 1 h	45	14.8	5.58 (SD = 3.34)	26.93 (SD = 8.75)
1–2 h	113	37.2	5.35 (SD = 3.27)	27.45 (SD = 10.86)
more than 2 h	66	21.7	6.97 (SD = 4.28)	31.27 (SD = 11.13)
does not watch daily	80	26.3	6.04 (SD = 4.16)	27.86 (SD = 9.47)
			*p* = 0.127 ***	*p* = 0.098 ***
Duration of Internet use before falling asleep				
does not use the internet	13	4.3	2.15 (SD = 2.15)	17.34 (SD = 8.53)
10–15 min	86	28.3	5.21 (SD = 3.41)	26.49 (SD = 9.92)
more than 30 min	121	39.8	5.79 (SD = 4.43)	28.44 (SD = 10.03)
more than 60 min	84	27.6	7.42 (SD = 4.24)	31.69 (SD = 10.04)
			*p* < 0.001 ***	F = 9.431; *p* < 0.001 **
Duration of daily PC, laptop, and tablet use				
max 1 h	8	2.6	6.50 (SD = 5.45)	25.63 (SD = 7.39)
1–2 h	39	12.8	4.97 (SD = 2.92)	23.54 (SD = 8.44)
more than 2 h	257	84.5	6.04 (SD = 3.85)	29.12 (SD = 10.51)
			F = 1.434; *p* = 0.24 **	*p* = 0.006 ***
Reading a book before falling asleep				
yes, almost every night	53	17.4	5.19 (SD = 3.45)	26.58 (SD = 8.26)
rarely	141	46.4	6.28 (SD = 4.10)	28.31 (SD = 11.07)
does not read books	110	36.2	5.79 (SD = 3.51)	29.15 (SD = 10.29)
			*p* = 0.209 ***	*p* = 0.351 ***
Coffee consumption				
yes, regularly	125	41.1	5.97 (SD = 3.58)	27.65 (SD = 9.40)
yes, occasionally	80	26.3	5.31 (SD = 3.27)	28.41 (SD = 10.83)
does not drink coffee	99	32.6	6.33 (SD = 4.38)	29.07 (SD = 11.13)
			*p* = 0.342 ***	*p* = 0.52 ***
Smoking habits				
yes	24	7.9	5.63 (SD = 4.01)	25.46 (SD = 11.77)
yes, occasionally	20	6.6	5.80 (SD = 4.10)	22.85 (SD = 7.84)
does not smoke	260	85.5	5.95 (SD = 3.79)	28.99 (SD = 10.25)
			F = 0.9; *p* = 0.914 **	F = 4.357; *p* = 0.014 **

There was a positive, moderate strength correlation (r = 0.548; r^2^ = 0.358) between AIS and PSS values (*p* = 0.001).

There are significant differences in the average values of the Perceived Stress Scale (PSS) based on the categories of the Athens Insomnia Scale (AIS) (F = 34.305; *p* = 0.001). Students without any sleep disorders had the lowest PSS score of 23.42 (SD = 9.63). The group of students with mild sleep disorders had a mean score of 31.33 (SD = 8.32), while those with moderate insomnia had a mean score of 38.62 (SD = 8.53). The group of students with a severe sleep disorder had a mean score of 28.66 (SD = 1.52) (Figure 1).

### 3.3. Possible Predictors of Insomnia

Students in relationships (6.39 ± 3.91) had significantly higher AIS scores than those who were single (5.32 ± 3.57) (*p* = 0.014). No significant differences were found between genders (*p* = 0.075) (Table 1).

Nurses appeared to be the most affected by sleep disorders (6.45 ± 3.81), while health visitors were the least affected (4.00 ± 2.32) (*p* = 0.044). Students at the beginning of their studies had better sleep quality. According to our results, students’ sleep quality seems to deteriorate continuously as their university years progress, although we did not find significant differences between the categories (*p* = 0.068) (Table 1).

We found a significant connection between the previous semester’s academic average and sleep quality (*p* < 0.001). Those with a moderate average (2.5–3.49) achieved significantly lower AIS scores (8.69 ± 4.19) compared to those with an academic average of 3.5 or above (5.55 ± 3.59) (*p* < 0.001) (Table 1).

The aspects of online education, specifically related to theoretical courses, did not significantly impact students’ sleep quality (*p* = 0.242). Almost three-quarters of students (74.3%) experience distraction during online lectures (Table 1).

Among students who regularly use their phones during online classes, the AIS average score (8.66 ± 4.47) significantly deviates in a negative direction compared to those who can pay attention throughout (4.40 ± 3.71) or get distracted (5.83 ± 3.51) (*p* = 0.001). Daily television and movie watching habits did not significantly influence sleep quality (*p* = 0.127). However, it is worth noting that students who spend more than 2 h daily on these activities had the worst average score (6.97 ± 4.28) (Table 2).

We analyzed the relationship between pre-sleep reading, phone usage, and sleep quality. The average AIS score was the lowest for those who do not use the internet before bedtime (2.15 ± 2.15). With the constant increase in pre-sleep internet usage, there was a consistent deterioration in sleep quality (*p* = 0.001). Pre-sleep reading habits did not significantly impact sleep quality (*p* = 0.209) (Table 2).

Students engaging in regular physical activity (5.28 ± 3.87) had a significantly lower AIS average score (*p* = 0.005). Coffee consumption and smoking habits did not significantly affect students’ sleep quality (*p* = 0.342; *p* = 0.914) (Table 2).

### 3.4. Possible Connections of Perceived Stress

Most students exhibit moderate stress, which may be attributed to the university as a stressor. The PSS score of male respondents (29.14 ± 10.13) was significantly lower than female respondents (*p* < 0.001). The perceived stress level of students in relationships (29.20 ± 11.05) was higher than single students (27.21 ± 9.35) (*p* = 0.960) (Table 1).

Significant differences were found among the PSS scores of different specializations (*p* = 0.001). Nursing students were the most affected (33.08 ± 10.56), while paramedics were the least affected (24.17 ± 10.65). Significant differences were also observed in the number of active semesters. Students in their seventh semester or beyond (32.84 ± 10.04) had significantly higher PSS scores (*p* = 0.045). Students with a moderate academic average (35.77 ± 10.59) also demonstrated significantly higher PSS scores compared to those with an excellent average (27.34 ± 9.94) (*p* < 0.001) (Table 1).

Students engaging in regular sports activities had significantly lower PSS scores than their peers (*p* < 0.001). Coffee consumption habits did not have a significant effect (*p* = 0.520), but smoking habits influenced the sample significantly. Students who smoke occasionally had significantly lower PSS scores (*p* = 0.014) (Table 2).

A higher PSS score was observed among students who prefer online lectures (32.23 ± 8.87) (*p* = 0.048). Those who could pay attention throughout online classes had significantly lower PSS scores (*p* < 0.001). Daily use of smart devices for more than 2 h increased student stress levels (29.12 ± 10.51) (*p* = 0.006) (Table 1 and Table 2).

Only 17.4% of students read a book before bedtime. However, these students achieved lower overall scores on the Perceived Stress Scale (26.58 ± 8.26) than those who did not read (29.15 ± 10.29). Most students spend more than 30 min on the internet before bedtime. As the pre-sleep internet usage increases, PSS scores also increase in a negative direction (*p* < 0.001) (Table 2).

**Table 3 healthcare-12-00075-t003:** The average values of the Athens Insomnia Scale and the Perceived Stress Scale, as well as the average values corresponding to the categories.

Measuring Tool	Category	N (%)	Mean (SD)
**Athens Insomnia Scale** **Mean = 5.91 (3.79)**	No sleep disorder	147 (48.4)	2.80 (1.63)
	Mild sleep disorder	118 (38.8)	7.50 (1.40)
	Moderate insomnia	35 (11.5)	12.68 (0.96)
	Severe sleep problems	3 (1.0)	17.11 (1.01)
**Perceives Stress Scale** **Mean 28.31 (10.35)**	Low stress level	66 (21.7)	14.15 (3.94)
	Moderate stress level	126 (41.4)	25.96 (3.25)
	High stress level	112 (36.8)	39.29 (4.93)

## 4. Discussion

The main goal of our research was to assess the sleep quality and stress levels of students during the online education period at our university. A total of 304 students completed our online questionnaire, including validated and self-designed questions, and the majority were female (n = 270). According to our findings, online education is associated with an increase in daily smart device usage, which correlates with higher stress levels. We identified a positive correlation between the duration of pre-sleep internet usage and AIS scores.

Students who were in relationships had a significantly higher average AIS score than those who were single (t = −2.470; *p* = 0.014). Being in a relationship often involves social activities and spending time together. Late-night outings or activities can lead to later bedtimes, potentially affecting the total duration and quality of sleep. Factors such as mattress comfort, room temperature, and noise levels may need to be negotiated to ensure both partners can sleep well [16,17].

As the results indicate, more than half of the students engage in regular physical activity, and the proportion of those who do not dedicate any time to physical activity in their free time is low. Among students with a medium grade point average (2.50–3.49), those who rarely/never exercise showed significantly higher AIS and PSS scores (*p* ≤ 0.05). Similar results were obtained in a study published by Fusz et al., examining the sleep quality of adolescents aged between 12 and 18. Students who performed better in school achieved lower AIS scores (r = −0.15; *p* = 0.034). Those who pay attention to healthy nutrition sleep better (4.73 ± 3.58) than those who do not pay attention at all (6.57 ± 4.25) (*p* < 0.001) [18]. Less physical activity had a significantly negative impact on both sleep quality and perceived stress. Insufficient physical activity can contribute to disruptions in circadian rhythms, affecting the natural sleep–wake cycle. Regular physical activity has been associated with better sleep efficiency and a more consistent sleep pattern. Students who engage in regular physical activity often report better coping mechanisms for stressors, leading to lower perceived stress levels [19,20].

There is a well-established connection between sleep quality and perceived stress, with poor sleep often exacerbating feelings of stress. Sleep deprivation or disturbances can lead to increased levels of stress [1,12,16]. Our research concurs with previous studies, indicating that there is a moderate, positive correlation between the AIS and PSS scores. This implies that higher levels of perceived stress are associated with poorer sleep quality among students.

Regular watching of a series/phone use during online classes, daily smart device usage exceeding 2 h, and more than 60 min of smart device use before bedtime also negatively influenced AIS and PSS scores (*p* ≤ 0.05). Similar correlations between sleep quality and AIS scores are observed in adolescents [7]. Students who do not use the internet before sleep much better (2.69 ± 2.34) than those who spend more than 60 min online before bedtime (5.62 ± 4.08) (*p* = 0.002). The constant connectivity and exposure to stimulating content on smart devices may contribute to heightened stress levels, further compromising the overall quality of sleep and well-being. As we delve into the era of pervasive technology, understanding the intricate relationship between smart device usage, sleep quality, and stress becomes essential for promoting healthier sleep habits and overall mental well-being [16,17,21].

Almost three-quarters of the students experience distractions during online lectures, which complicates the later mastery of the material. Most respondents use their smart devices for more than 2 h daily, which is attributed partly to online education and partly to the internet dependence developed during the COVID-19 pandemic. Online lecturing can be susceptible to attention distraction, as students may face a myriad of digital interruptions, such as social media notifications and emails, competing for their focus during virtual classes. Additionally, the absence of physical presence and direct monitoring in online environments may make it easier for students to succumb to multitasking, diminishing their ability to fully engage with the lecture content and impacting overall attention retention [2,7,22].

Smart devices emit blue light, which can suppress melatonin production and disrupt circadian rhythms. Using these devices before bedtime may interfere with the natural sleep–wake cycle and contribute to poor sleep quality. Excessive screen time, especially close to bedtime, has been associated with insomnia and difficulty falling asleep. Students who spend more time on their smart devices in the evening may experience disrupted sleep patterns. Some smart devices offer features like night mode or blue light filters that reduce the impact of blue light on melatonin production. These features may be beneficial for users who need to use devices in the evening [21,23,24].

To improve sleep quality, adhering to sleep hygiene rules is essential, as sleep problems contribute to various illnesses and disorders [3]. The results indicate that only 17.4% of students regularly read before bedtime, while most spend more than 30 min online. Background noise and light during sleep increase the feeling of morning fatigue. A negative correlation was also found in adolescents between adherence to sleep hygiene rules and AIS scores (*p* < 0.001). Only 18% of students read a book almost regularly before bedtime, while the majority (47%) do so rarely, and 35% do not [18].

Understanding the link between sleep and cognitive function can motivate students to prioritize their sleep. Universities can organize workshops or provide resources on sleep hygiene. These sessions can cover topics such as creating a sleep-friendly environment, establishing good sleep habits, and addressing common sleep challenges. Universities should offer counseling services to students experiencing stress, anxiety, or other mental health challenges that may impact sleep [22,25]. The impact of reading before sleep can vary among individuals. Some may find it extremely helpful, while others may prefer different pre-sleep activities. While recreational reading can be beneficial, students should be cautious about reading academic material before bedtime. Studying or reading complex material might stimulate the brain and hinder the ability to unwind [26,27].

Regarding our study, nursing, physiotherapy, and radiography students proved to be the most affected in terms of insomnia and perceived stress (*p* ≤ 0.05). The sleep quality and perceived stress among university students pursuing healthcare professions can be influenced by a variety of factors unique to the demands of their academic and clinical training. A heavy academic workload can contribute to stress and potentially impact sleep quality. The demands of hands-on training can be physically and mentally exhausting, affecting both sleep and stress levels. Some healthcare professions, such as nursing, involve shift work. Irregular schedules and night shifts can disrupt circadian rhythms, making it challenging for students to maintain consistent sleep patterns [28,29,30,31].

The elevated scores on the AIS and PSS among university students highlight potential clinical implications for mental health professionals. These findings suggest a need for targeted interventions and support programs addressing sleep quality and stress management within the university population. Implementing strategies to improve sleep hygiene and coping mechanisms may be crucial in promoting overall well-being and academic success among students. Additionally, educational campaigns and peer support initiatives can contribute to creating a campus culture that prioritizes mental well-being.

## 5. Limitations

Despite the valuable insights gained from this study, it is important to acknowledge several limitations. Firstly, the research relied on self-reported data, introducing the possibility of response bias and subjective interpretation of variables, such as sleep quality and stress levels. Additionally, the study’s cross-sectional nature limits the establishment of causal relationships among the examined factors. The study’s focus on a single institution and specific student population may also impact the generalizability of findings to broader contexts. Incorporating professional development courses that teach effective coping strategies, resilience, and mindfulness can help students navigate the challenges of their chosen healthcare professions. Future research employing diverse methodologies and more prominent and representative samples would contribute to a more comprehensive understanding of the complex interplay between smart device use, sleep quality, and perceived stress in the context of online education.

## 6. Conclusions

In conclusion, this study conducted among university students at the Health Sciences Faculty investigated the intricate relationship between various factors, including smart device usage, sleep quality, and perceived stress during online education. The findings revealed significant associations, highlighting that excessive smart device use was linked to poorer sleep quality and elevated stress levels among the surveyed students. These outcomes underscore the importance of implementing targeted interventions and recommendations to mitigate the adverse effects of these factors on the well-being of the student population, particularly in the context of online education.

## Figures and Tables

**Figure 1 healthcare-12-00075-f001:**
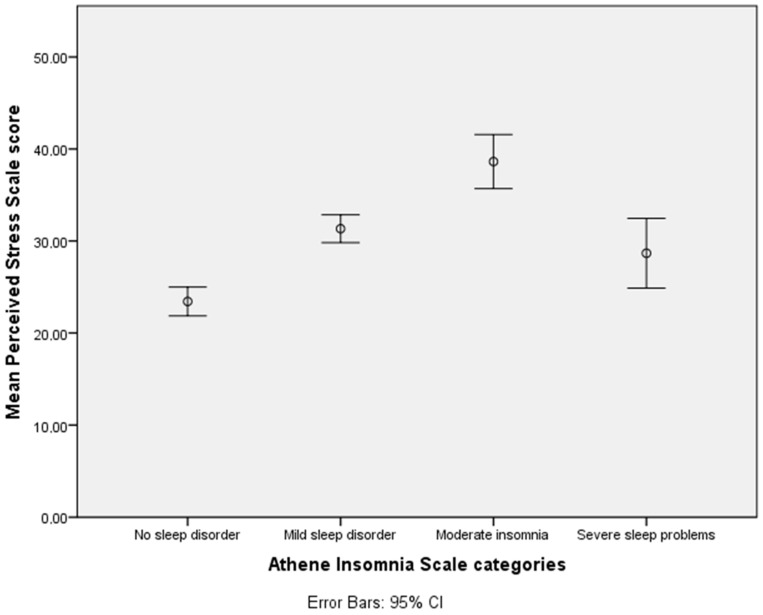
The relationship of the average Perceived Stress Scale values corresponding to the categories of the Athens Insomnia Scale.

**Table 1 healthcare-12-00075-t001:** The relationship between university education characteristics, exercise habits, online learning experiences, and the quality of sleep and perceived stress (* independent sample *t*-test; ** analysis of variance; *** Kruskal–Wallis).

	n	%	AIS	PSS
Gender				
female	270	88.8	6.05 (SD = 3.71)	29.14 (SD = 10.13)
male	34	11.2	4.82 (SD = 4.29)	21.74 (SD = 9.87)
			t = 1.785; *p* = 0.075 *	t = 4.028; *p* < 0.001 *
Family status				
single	136	44.7	5.32 (SD = 3.57)	27.21 (SD = 9.35)
in a relationship	168	55.3	6.39 (SD = 3.91)	29.20 (SD = 11.05)
			t = −2.47; *p* = 0.014 *	t = −1.671; *p* = 0.96 *
Specialty				
nurse	53	17.4	6.45 (SD = 3.81)	33.08 (SD = 10.56)
physiotherapist	121	39.8	6.20 (SD = 3.60)	28.57 (SD = 9.63)
paramedic	30	9.9	5.17 (SD = 4.56)	24.17 (SD = 10.65)
radiographer	51	16.8	6.44 (SD = 4.11)	27.22 (SD = 9.85)
laboratory analytic	25	8.2	5.08 (SD = 3.65)	27.28 (SD = 10.75)
health visitor	24	7.9	4.00 (SD = 2.32)	25.08 (SD = 10.58)
			*p* = 0.044 ***	F = 4.045; *p* = 0.001 **
University semester				
1–2	113	37.2	5.62 (SD = 3.47)	28.30 (SD = 10.46)
3–4	103	33.9	6.06 (SD = 4.53)	27.30 (SD = 11.28)
5–6	56	18.4	5.91 (SD = 2.76)	27.61 (SD = 7.76)
7+	32	10.5	6.50 (SD = 3.95)	32.84 (SD = 10.04)
			*p* = 0.68 ***	*p* = 0.045 ***
Average of the previous semester				
medium	35	11.5	8.69 (SD = 4.19)	35.77 (SD = 10.59)
excellent	269	88.5	5.55 (SD = 3.59)	27.34 (SD = 9.94)
			t = 4.757; *p* < 0.001 *	t = 4.685; *p* < 0.001 *
Frequency of sports				
regularly	165	54.3	5.28 (SD = 3.87)	25.49 (SD = 10.09)
rarely	123	40.5	6.73 (SD = 3.53)	31.84 (SD = 9.30)
does not play sports	16	5.3	6.13 (SD = 4.03)	30.31 (SD = 12.44)
			F = 5.315; *p* = 0.005 **	F = 14.789; *p* < 0.001 **
Online Education				
good in theory	158	52	6.03 (SD = 3.89)	28.43 (SD = 10.48)
suitable in every aspect	31	10.2	6.77 (SD = 3.58)	32.23 (SD = 8.87)
personal is better	115	37.8	5.53 (SD = 3.70)	27.10 (SD = 10.35)
			F = 1.425; *p* = 0.242 **	F = 3.06; *p* = 0.048 **
Regarding online education				
getting distracted	226	74.3	5.83 (SD = 3.51)	28.39 (SD = 9.91)
paying attention the whole time	46	15.1	4.40 (SD = 3.71)	22.67 (SD = 9.45)
usually on the phone/watching a series	32	10.5	8.66 (SD = 4.47)	35.84 (SD = 9.97)
			*p* < 0.001 ***	F = 16.901; *p* < 0.001 **

## Data Availability

The datasets generated and/or analyzed during the current study are not publicly available due to privacy, confidentiality, and other restrictions.

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
