# Peer review of "Sleep Quality and Perceived Stress among Health Science Students during Online Education—A Single Institution Study"

_healthcare, 2023, doi:10.3390/healthcare12010075_

Round 1

Reviewer 1 Report

Comments and Suggestions for Authors

The paper by Busa et al. reports the results of a cross-sectional study investigating sleep quality and perceived stress levels among students of a  University of Health Sciences. Although interesting, there are some major concerns.

Major comments

  1. Tables 1 and 2 are missing. Therefore it is difficult to review the results.
  2. It is advisable to include information on the reliability (internal consistency) and validity data of the standardized instruments AIS and PSS for the target population. Original validation studies might be referenced for these two questionnaires.
  3. The statistical analysis is poor and crucial analysis, such as adjusting for confounding factors, is absent.
  4. Did the survey period coincide with an examination period?
  5. What was the percentage of participants who responded to the study?
  6. Additionally, there is an issue of females being overly represented. The authors should comment on that.
  7. Discussion section needs improvements. Clinical implication of the study should be presented more clearly to the reader.

Author Response

Dear Reviewer 1!

Thank you sincerely for your thoughtful and constructive feedback on our work. Your insights have been invaluable in refining and improving the quality of our research. We appreciate your time and expertise. 

Regarding your comments:

Tables 1 and 2 are missing. Therefore it is difficult to review the results. – INSERTED INTO MANUSCRIPT

It is advisable to include information on the reliability (internal consistency) and validity data of the standardized instruments AIS and PSS for the target population. Original validation studies might be referenced for these two questionnaires. – CRONBACH ALPHA’S INCLUDED. REGARDING VALIDATION WE CITED THE ORIGINAL (VALIDATED) REFERENCES.

The statistical analysis is poor and crucial analysis, such as adjusting for confounding factors, is absent. – SOME STATISTICAL ANALYZISES HAVE BEEN INSERTED.

Did the survey period coincide with an examination period? – WE PAID DIRECT ATTENTION, THERE WAS NO COINCIDE WITH AN EXAMINATION PERIOD.

What was the percentage of participants who responded to the study?  - INSERTED

Additionally, there is an issue of females being overly represented. The authors should comment on that. – THE HUNGARIAN HEALTHCARE SYSTEM IS CHARACTERIZED BY FEMALE OVERREPRESENTATION, ESPECIALLY IN NON-DOCTORAL PROFESSIONS. THIS OVERREPRESENTATION IS ALSO PRESENT AMONG UNIVERSITY STUDENTS.

Discussion section needs improvements. Clinical implication of the study should be presented more clearly to the reader. – WE INSERTED SOME NEW THOUGHTS.

Reviewer 2 Report

Comments and Suggestions for Authors

I have the following comments:

1. Abstract: Fine.

2. Introduction: The information is generally fine, but why doesn't this MS explore further what factors are associated with students' sleep quality?

3. Materials and Methods

3.1. Included power analysis

3.2. In comment 2, the authors can use AIS to identify different insomnia groups and associate them with students' demographics, stress, habits, etc.

4. Results:

4.1. Tables and Figures have to be included in this MS instead of supplementary materials.

4.2. what is the response rate? Will the response rate affect the generalizability of the results, and do we need to include this in the limitation?

5. Discussion: Fine

Author Response

Dear Reviewer 2!

We extend our sincere gratitude for your thorough review of our work. Your constructive feedback and valuable suggestions have significantly enhanced the quality of our research. We appreciate your time, expertise, and thoughtful contributions to the refinement of our manuscript.

Regarding your comments:

1. Abstract: Fine. – THANK YOU

2. Introduction: The information is generally fine, but why doesn't this MS explore further what factors are associated with students' sleep quality? - INSERTED

3. Materials and Methods

3.1. Included power analysis – WE HAVE RE-DESIGNED

3.2. In comment 2, the authors can use AIS to identify different insomnia groups and associate them with students' demographics, stress, habits, etc. – THE AIS AND PSS GROUPS WERE CALCULATED, HOWEVER THE NUMBER OF STUDENTS AT AIS „SEVERE SLEEP PROBLEMS” WAS TOO LOW. THEREFORE THE ASSOCIATIONS OF THE MENTIONED APPROACH WOULD BE INCORRECT.

4. Results:

4.1. Tables and Figures have to be included in this MS instead of supplementary materials.- INCLUDED

4.2. what is the response rate? Will the response rate affect the generalizability of the results, and do we need to include this in the limitation? - INCLUDED

5. Discussion: Fine – THANK YOU

Reviewer 3 Report

Comments and Suggestions for Authors

I read and analyzed the manuscript Sleep Quality and Perceived Stress Among Health Science Students During Online Education – Single Institution Study in detail with satisfaction.

The topic is certainly up-to-date because the well-being of all students, especially health sciences students who need to care for others soon, is important to us.

However, the manuscript has numerous weaknesses, and there is no significant wider impact, except locally, unless major corrections are made.

Introduction

In the Introduction, the authors analyze the influence of sleep hygiene on sleep quality, and in just two paragraphs, they refer to perceived stress and online education. Taking into account the title of the manuscript, in the Introduction, more attention should be paid to online education regarding its impact on sleep quality and perceived stress. There is a lack of information about this and why it is important for health science students. Namely, the author's main focus in the introductory part is the factors affecting the quality of sleep in terms of sleep hygiene. Whether and why students' sleep hygiene would worsen during online education remains unclear.

Materials and Methods

The authors start the methodology with ethical considerations instead of the research procedure. The sample size and how the authors defined it is unclear. Specifying the sample and how they determined the sample size is necessary. Also, it was stated that all students (and their number is unknown) received the questionnaire via email and that "something was shared" via social networks. Whether the questionnaire was shared via social media or research information is unclear.

The description of the instrument lacks information on whether it is publicly available or whether consent is required for their use and whether they have received it in that case. There is a lack of information on how the questionnaires were translated into Hungarian and the reliability value in the given context.

It is necessary to make a subtitle of ethical considerations as was done for statistical analysis.

Results

Listing all the data more than presented in a table is unnecessary. The tables lack a legend based on which readers would know whether parametric or non-parametric statistics were used. Also, line 173 states, "We found a correlation between …." And correlation tests are not listed or shown.

All in all, many burdensome texts with results already well-known in the literature exist. It would be good if the authors checked the correlation between sleep quality and stress and trusted the predictor variables with regression analysis. Analyzes carried out and presented in this way do not significantly impact the obtained results, except at the local level.

Discussion

Considering the suggestions in the previous parts of the manuscript and the discussion, they notice similar shortcomings.

General recommendation

Explain online education (when and how much is carried out). Perform additional statistical analyses, a focused introduction, and an in-depth discussion.

Author Response

Dear Reviewer 3,

Thank you immensely for your insightful and detailed review of our manuscript. Your thoughtful comments and suggestions have proven invaluable in strengthening the overall quality of our work. We are genuinely appreciative of your time and expertise in contributing to the refinement of our research.

Regarding your comments:

In the Introduction, the authors analyze the influence of sleep hygiene on sleep quality, and in just two paragraphs, they refer to perceived stress and online education. Taking into account the title of the manuscript, in the Introduction, more attention should be paid to online education regarding its impact on sleep quality and perceived stress. There is a lack of information about this and why it is important for health science students. Namely, the author's main focus in the introductory part is the factors affecting the quality of sleep in terms of sleep hygiene. Whether and why students' sleep hygiene would worsen during online education remains unclear. – WE HAVE REWROTE SOME SECTIONS AND ADDED SOME ASPECTS

Materials and Methods

The authors start the methodology with ethical considerations instead of the research procedure. The sample size and how the authors defined it is unclear. Specifying the sample and how they determined the sample size is necessary. Also, it was stated that all students (and their number is unknown) received the questionnaire via email and that "something was shared" via social networks. Whether the questionnaire was shared via social media or research information is unclear. – WE HAVE TRIED TO EXPLAIN EVERYTHING YOU MENTIONED

The description of the instrument lacks information on whether it is publicly available or whether consent is required for their use and whether they have received it in that case. There is a lack of information on how the questionnaires were translated into Hungarian and the reliability value in the given context. – WE HAVE MADE CORRECTIONS

It is necessary to make a subtitle of ethical considerations as was done for statistical analysis. – THANK YOU, WE HAVE DONE IT

Results

Listing all the data more than presented in a table is unnecessary. The tables lack a legend based on which readers would know whether parametric or non-parametric statistics were used. Also, line 173 states, "We found a correlation between …." And correlation tests are not listed or shown. – WE HAVE ADDED LEGENDS AND CORRECTED OUR MISTAKE

All in all, many burdensome texts with results already well-known in the literature exist. It would be good if the authors checked the correlation between sleep quality and stress and trusted the predictor variables with regression analysis. Analyzes carried out and presented in this way do not significantly impact the obtained results, except at the local level. – WE HAVE MADE THE ANALYZIS AND INSERTED THE RESULTS AND A NEW TABLE AND FIGURE INTO MANUSCRIPT

Discussion

Considering the suggestions in the previous parts of the manuscript and the discussion, they notice similar shortcomings. – WE HAVE MADE SOME NEW EXPLANATIONS

General recommendation

Explain online education (when and how much is carried out). Perform additional statistical analyses, a focused introduction, and an in-depth discussion. – WE TRIED OUR BEST, THANK YOU ONCE AGAIN

Round 2

Reviewer 2 Report

Comments and Suggestions for Authors

This revised MS has improved a lot, especially since the authors have addressed all the comments from the reviewers on this revised manuscript. Based on the current version, I will accept publishing it to Healthcare unless other reviewers or the editor have additional comments requiring further revisions. Well done!

Reviewer 3 Report

Comments and Suggestions for Authors
